# Evaluating the Diagnostic Performance of PET/MR Versus CECT in Determining Resectability in Ovarian Cancer

**DOI:** 10.3390/cancers17162612

**Published:** 2025-08-09

**Authors:** Mayur Virarkar, Sanaz Javadi, Aatiqah Aziz, Jia Sun, Revathy Iyer, Dhakshinamoorthy Ganeshan, Taher Dauod, Priya Bhosale

**Affiliations:** 1Department of Radiology, University of Florida, Jacksonville, FL 32209, USA; 2Department of Diagnostic Radiology, MD Anderson Cancer Center, Houston, TX 77030, USA; sanaz.javadi@mdanderson.org (S.J.); aaziz2@mdanderson.org (A.A.); riyer@mdanderson.org (R.I.); dganeshan@mdanderson.org (D.G.); tedaoud@mdanderson.org (T.D.); priya.bhosale@mdanderson.org (P.B.); 3Department of Biostatistics, MD Anderson Cancer Center, Houston, TX 77030, USA; jsun9@mdanderson.org

**Keywords:** PET/MR, CECT, ovarian, specificity, accuracy, NPV

## Abstract

The aim of this research is to evaluate and compare the diagnostic performance of hybrid PET/MR imaging and contrast-enhanced computed tomography (CECT) in preoperative assessment of resectability among patients with ovarian cancer. Existing imaging modalities exhibit notable variability in sensitivity and specificity for detecting peritoneal and extraperitoneal disease, particularly within challenging anatomical regions. By conducting a systematic, site-based comparison of PET/MR and CECT findings against intraoperative and pathological standards, we aim to delineate the relative strengths and limitations of these modalities. The results are intended to delineate whether the integration of functional and anatomical imaging via PET/MR offers a significant advantage over standard multi-detector CT and potentially enhances treatment approaches in advanced ovarian cancer.

## 1. Introduction

Ovarian cancer is one of the most lethal gynecologic malignancies [1]. Accurate staging and comprehensive assessment of disease extent are pivotal in guiding management strategies and optimizing patient outcomes. While diagnostic laparoscopy is considered the gold standard for lesion detection and surgical planning, it is invasive and resource-intensive, necessitating the evaluation of non-invasive imaging modalities that can offer comparable diagnostic accuracy [2].

Traditionally, contrast-enhanced computed tomography (CECT) has been the imaging cornerstone in evaluating ovarian cancer due to its widespread availability and established role in oncologic imaging [3]. It has been used to predict suboptimal resection, with a sensitivity ranging from 28% to 87% and a specificity ranging from 43% to 100% [4]. One recent study showed high predictability of suboptimal debulking using CT [5]. However, emerging hybrid imaging techniques like positron emission tomography/magnetic resonance imaging (PET/MR) have shown promise in oncologic imaging, combining the functional capabilities of PET with the superior soft-tissue contrast of MR [6]. This dual capability positions PET/MR as a potentially superior modality for detecting peritoneal and small-volume disease, critical for accurate staging and treatment planning in ovarian cancer (Figure 1).

The primary objective of this study is to compare the diagnostic accuracy of PET/MR and contrast-enhanced CT in patients with advanced-stage serous ovarian cancer or those with a high clinical suspicion of ovarian malignancy undergoing a planned upfront debulking procedure. Using diagnostic laparoscopy as the gold standard, this study seeks to evaluate the performance of these imaging modalities for lesion detection, aiming to identify the strengths and limitations of each technique. By advancing our understanding of imaging efficacy in this context, this research could inform clinical decision-making, improve non-invasive diagnostic approaches for patients with ovarian cancer, and help predict optimal debulking.

## 2. Methods

### 2.1. Patient Data Collection

All patients enrolled in the study underwent PET/MRI and CECT scans. The CECT was sometimes performed before enrollment, but both imaging studies occurred within 17.5 days (±12 days) before the scheduled laparoscopy or surgical cytoreduction. Patients who did not undergo immediate surgical cytoreduction were reimaged with PET/MRI and CECT after completing chemotherapy. The PET/MR was performed 3.9 days (±2.5) for patients who underwent optimal debulking surgery, and the second PET/MR was performed 106 days (±29.4) before surgery in patients who underwent neoadjuvant platinum-based chemotherapy. All patients had serous carcinoma, with the following breakdown: 13 high-grade serous, 1 granulosa cell tumor, 1 mucinous adenocarcinoma, 1 Müllerian high-grade, 1 clear cell carcinoma, and 2 peritoneal cancers. Most patients presented with advanced-stage disease (FIGO stage III or IV), as our inclusion criteria focused on those undergoing evaluation for cytoreductive surgery. The mean age was 49.1 years (39.6–75.1 years).

### 2.2. Surgical and Imaging Assessments

All participants underwent diagnostic laparoscopy to assess tumor location and the potential for cytoreduction. The size and location of tumor implants were meticulously documented and scored based on a predefined system that evaluated various anatomical sites.

The scoring methodology systematically evaluated tumor spread across multiple anatomical sites by assigning specific point values, including small bowel mesentery (1 point), mesocolon (1 point), hepatic parenchyma (1 point), hepatic hilum or surface involvement (1 point), supracolic and infracolic omental extensions (1 point), gastrosplenic ligament involvement (1 point), splenic surface involvement (1 point), diaphragmatic involvement (1 point), peritoneal thickening (1 point), macroscopic peritoneal implants ≥2 cm (1 point), miliary visceral peritoneum implants (1 point), massive ascites (1 point), suprarenal para-aortic lymph nodes ≥1 cm (1 point), anterior diaphragmatic lymph nodes (1 point), and pleural effusion (1 point). Pelvic sidewall involvement and/or hydroureter were assigned 0 points based on the Fagotti scoring system (Table 1). This scoring system enabled the radiologists and gynecologic oncologists to assess the extent of the disease comprehensively, evaluate the feasibility of surgical cytoreduction, and document tumor implant characteristics systematically for treatment planning, prognostic assessment, and research purposes. A score of >8 resulted in the patient receiving neoadjuvant chemotherapy. This scoring system was applied to all patients who underwent upfront surgical debulking and neoadjuvant chemotherapy.

### 2.3. Materials and Methods

All studies were performed on the GE 3.0T PET/MR HD 750w scanner (GE Healthcare, Waukesha, WI, USA). The same technologists performed the study for each patient. The patient was asked to void before the exam. The MRI portion of the PETM included a fat-saturated T1 pre-contrast, T1 fastest post-contrast, DWI, and T2 FSE fat-sat sequence through the chest, abdomen, and pelvis. Small FOV T2 fat-sat, Sagittal and axial FOV DWI, and 3D Sagittal post-contrast images were obtained through the pelvis. All patients received ultrasound gel in the vagina before imaging. Gadavist (Gadavist, Bayer Healthcare Pharmaceuticals, Leverkusen, Germany) was used as a contrast agent. PET/MRI data were acquired in LIST mode and reformatted into multiple image frames for analysis. The imaging data were reconstructed using OSEM with TOF and PSF techniques. The images were sent to the MIM workstation 7.1.4 and were interpreted by a radiologist specializing in gynecologic imaging for the past 20 years (PB). The SUV was documented. The ADC values were reported on the PACs workstation. The lowest ADC value and the highest SUV value were documented , including the primary tumor and the metastatic disease. For lymph node assessment, FDG-avid lymph nodes were considered positive for metastases, regardless of whether the size of CT lymph nodes >1 cm in the short axis, which were considered metastatic.

CT imaging was performed through the chest, abdomen, and pelvis. The images were obtained in the PV phase of contrast enhancement. Sagittal and coronal reformats were obtained. Omnipaque 350 (GE Healthcare, Waukesha, WI, USA) was a contrast agent. The CT images were interpreted on a PACS workstation by a radiologist with 10 years of experience (SJ), and the PET/MR images were interpreted by a radiologist (PB) with 20 years of experience and specializes in gynecologic oncology. Both radiologists were ignorant of the other radiologist’s interpretation. The data was collected on an Excel sheet by a research assistant (AA) who worked with the interpreting radiologist. Surgical pathology was considered the gold standard. Correlation was available only for the abdominal and pelvic disease.

All patients underwent laparoscopic evaluation despite the disease seen on imaging findings per protocol. They independently scored the implants, and the decision for neoadjuvant therapy was based on laparoscopic findings. Twelve patients who were deemed to be resectable during laparoscopy underwent upfront optimal debulking surgery.

Resectability in ovarian cancer is determined through clinical, imaging, and surgical assessments. PET/MR and CT were used to evaluate disease spread to critical anatomical sites (e.g., small bowel mesentery, hepatic surface, lymph nodes). We applied the Fagotti scoring system, which assigns points to disease involvement at specific sites. A score >8 was considered predictive of unresectability and guides the decision for neoadjuvant chemotherapy. All patients underwent diagnostic laparoscopy, which remains the gold standard for assessing resectability and guiding management.

### 2.4. Inclusion/Exclusion Criteria

Patients enrolled in the prospective trial were aged over 18 years, had ovarian cancer or were highly suspected of having it, and were deemed suitable for laparoscopic evaluation for resectability. Inclusion also required an estimated glomerular filtration rate (eGFR) greater than 30. Exclusion criteria included a known allergy to gadolinium, eGFR below 30, presence of a cardiac pacemaker, age under 18 years, body weight over 180 kg, blood glucose levels above 200 mg/dL, and pregnancy.

### 2.5. Statistical Analysis

Lesion detection was summarized using frequencies and percentages by modality. Sites of metastases were summarized using frequencies and percentages. Metastases diagnosed by PET/MR or CT were cross-tabulated. Statistical analysis was performed using R (version 4.3.1, R Development Core Team).

Overall Survival (OS) is defined as the duration from the date of diagnosis to the date of death or loss to follow-up. Recurrence-Free Survival (RFS) is defined as the duration from the date of diagnosis to the recurrent event or death or to loss to follow-up. OS and RFS were estimated using the Kaplan–Meier method.

Cox regression was used to associate the candidate variables (SUV and ADC) with OS. All tests were two-sided, and *p*-values of 0.05 or less were considered statistically significant. Statistical analysis was performed using R (version 4.3.1, R Development Core Team).

## 3. Results

The average age of patients was 49.1 years (max 75.1, and min 39.6). Thirteen patients had serous cancers. One patient had an adult granulosa cell tumor, one patient had moderately differentiated adenocarcinoma with mucinous differentiation, one patient had high-grade mullerian cancer, one patient had clear cell cancer, and two patients had peritoneal cancer. Six patients were deemed unresectable either based on scoring >8 on imaging or laparoscopic assessment or based on PET/MR, and they received Taxol and carboplatin. Of six patients who received chemotherapy, three patients (50%) experienced management changes based on PET/MR findings and based on identification of metastatic supraclavicular and mediastinal adenopathy (anterior diaphragmatic and internal mammary) FDG-avid adenopathy. The remaining three showed extensive disease on both imaging modalities, specifically on laparoscopy, and one patient has had liver infiltration, one patient had lesser sac disease, and one patient had extensive diaphragmatic infiltration. Four patients had a score of ≥8 both on CT and the PET/MR, which correlated with the laparascopy and received neoadjuvant therapy.

On a per-patient basis, PET/MR scored higher than CT. Two patients scored <8 but had adenopathy outside the abdomen, and one had a perihepatic infiltrating large implant detected on PET/MR. One patient with a CT score of 8 and a PET/MR score of 7 underwent upfront debulking surgery (positive anterior diaphragmatic adenopathy was the trigger to give chemotherapy). None of the patients had hydroureter, hydronephrosis, or pelvic sidewall involvement.

### Metastases Detected on CT and PET/MR

PET/MR imaging demonstrated moderate sensitivity in detecting metastases across anatomical sites (Table 2). Supracolic omental disease was seen in 36.8% of cases (7/19), and infracolic omental disease in 42.1% (8/19). PET/MR reported small bowel mesenteric disease in 5.3% of patients (1/19), while perihepatic disease was found in 26.3% (5/19). On PET/MR, colon involvement and pelvic adenopathy were noted in 26.3% (5/19) and 10.5% (2/19) of cases, respectively.

On CT, infracolic omental disease was reported in 89.5% of patients (17/19), and supracolic omental disease in 52.6% (10/19). CT identified small bowel mesenteric disease in 36.8% (7/19) and perihepatic disease in 36.8% (7/19). CT detected colon involvement in 31.6% (6/19) and pelvic adenopathy in 5.3% (1/19).

PET/MR detected perihepatic metastases in one of the three patients and overestimated in four patients, with a PPV of 20% and a NPV of 86% (Table 3). PET/MR correctly identified one of the two patients for small bowel mesenteric disease. PET/MR overestimated small bowel mesenteric involvement in 1 patient, with a PPV of 100% and a NPV of 89%. It correctly identified small bowel involvement in 1 patient and overestimated in 1 patient, with a PPV of 50%, and a NPV of 100%. The missed colon involvement by PET/MR in 2 of 3 patients, overestimated colonic involvement in 4 patients, and had a PPV of 20% and a NPV of 86% . Also, the missed metastatic adenopathy by PET/MR in 1 of 4 patients had a PPV of 100% and a NPV of 94%.  CT identified two of the three patients correctly with perihepatic disease and overestimated in five patients, with a PPV of 29% and a NPV of 92%. CT correctly identified three patients with small bowel mesenteric disease and overestimated four patients, with a PPV of 43% and a NPV of 100%. CT missed one patient with small bowel serosal disease and overestimated two patients, with a PPV of 0% and a NPV of 94%. CT overestimated three patients with colonic involvement and correctly identified three patients with colonic involvement, with a PPV of 50% and a NPV of 100%. Of the four patients with adenopathy, one was correctly identified and three were missed. CT overestimated adenopathy in 2 patients, had a PPV of 33%, and a NPV of 81%.

Compared to the gold standard of pathology, PET/MR’s sensitivity for small bowel mesenteric disease was 33% (1/3), with a specificity of 100% (16/16). CT achieved perfect sensitivity (100%) for the same site but a lower specificity of 75% (12/16). For perihepatic disease, PET/MR had a sensitivity of 33% (1/3) and specificity of 75% (12/16), while CT showed a sensitivity of 67% (2/3) and a specificity of 69% (11/16). For colon involvement, PET/MR sensitivity was 33% (1/3) and specificity was at 75%, while CT had a sensitivity of 100% and specificity at 81% (Table 4). These findings affirm PET/MR’s strength in avoiding false positives and CT’s advantage in catching more true positives. The overall survival was 93% at 1 year, 79% at 2 years, and 79% at 3 years. The recurrence-free survival at 1 year was 81.2%, 45.1% at 2 years, and 33.8% at 3 years. Nine patients recurred post-treatment.

The pretreatment ADC and SUV did not impact overall survival (*p* = 0.394 and *p* = 0.774) or recurrence-free survival (*p* = 0.287 and *p* = 0.443).

## 4. Discussion

This study represents one of the first direct comparisons between PET/MR and CT for evaluating suboptimal resectability in ovarian cancer. Both modalities demonstrated equivalent diagnostic accuracy (68%) in detecting perihepatic disease. PET/MR exhibited superior performance for small bowel involvement with an accuracy of 95%, compared to 84% for CT. In contrast, CT demonstrated higher accuracy for colonic participation (84%) versus PET/MR (74%).

These findings are consistent with Tsuyoshi et al., who reported higher diagnostic accuracy of PET/MRI (92.5%) compared to contrast-enhanced MRI (80.6%) and superior performance in M staging (100% vs. 30.8%), while maintaining comparable accuracy for T and N staging [7]. In our cohort, PET/MR outperformed CT in detecting metastatic disease, with 82% and 72.9% accuracy rates, respectively.

In our cohort, PET/MR demonstrated 88.6% negative predictive value (NPV) compared to CT’s 91.5% NPV. However, both modalities had limitations in detecting certain types of metastatic disease. Our study found that PET/MR missed 1 of 4 patients with metastatic adenopathy and had 33% sensitivity for perihepatic disease. CT missed 3 of 4 patients with adenopathy but showed 100% sensitivity for small bowel mesenteric disease. For patients with negative imaging but high clinical suspicion, diagnostic laparoscopy remains the gold standard, as demonstrated by our protocol, where all patients underwent laparoscopic evaluation regardless of imaging findings. Clinical management should consider the probability of false negatives based on the specific anatomical sites and tumor characteristics, with consideration for diagnostic laparoscopy in cases where imaging is discordant with clinical suspicion.

A meta-analysis evaluating contrast-enhanced CT versus PET/CT in stage III ovarian cancer across 15 studies (n = 918) found no statistically significant difference in sensitivity between CT (82%) and PET/CT (87%) (*p* = 0.29). However, PET/CT demonstrated significantly higher specificity (90% vs. 72%, *p* < 0.01) [8]. Similarly, Tsili et al. conducted a systematic review and meta-analysis comparing MDCT, MRI, and FDG PET/CT for peritoneal metastases detection. MRI (82.7%) and FDG PET/CT (93.7%) outperformed MDCT (79.7%) in sensitivity on a per-patient basis. On a per-region basis, MRI showed the highest sensitivity (92.6%) compared to MDCT (70.1%) and FDG PET/CT (58.3%) [3]. These findings suggest that while PET/CT enhances diagnostic specificity, MRI may be the most sensitive modality for comprehensive peritoneal disease assessment due to its superior soft-tissue contrast, and incorporating PET and MRI can be beneficial for detecting metastatic disease. Khiewvan et al. [9] emphasized the diagnostic strength of PET/CT (sensitivity and specificity of 92–95%) in detecting metastases and recurrence, influencing treatment in up to 60% of cases. They also highlighted the emerging role of PET/MRI and the potential application of non-FDG PET tracers. In our study, PET/MR demonstrated a sensitivity of 100% for small bowel serosal involvement, compared to 75% for CT. This aligns with findings from Xi et al., who showed PET/MR had higher sensitivity than PET/CT in detecting gastrointestinal invasion (75% vs. 50%) %. [10]. However, systematic reviews by Suppiah et al. [11] and Li et al. [12] have further validated the performance of PET/CT, reporting a pooled sensitivity of 92%, specificity of 88%, and an AUC of 0.96 for detecting metastatic disease. MRI achieved high sensitivity (0.95) but comparatively lower specificity (0.81) [11].

In our data, PET/MR demonstrated higher specificity for perihepatic disease (75%) compared to CT (69%), albeit lower than previously reported by Xi et al. (PET/CT: 90.9%; PET/MR: 81.8%) [10]. Accurate identification of perihepatic disease is clinically significant, as the size and extent of hepatic implants directly impact surgical planning. MRI may provide better contrast resolution in identifying perihepatic disease.

PET/MR also showed superior performance in detecting metastatic adenopathy with sensitivity and specificity of 75% and 100%, respectively, compared to CT (25% and 87%). However, these values are slightly lower than the 100% sensitivity and specificity reported by Xi et al. for both PET/CT and PET/MR [10]. The ability of PET/MR to assess both metabolic activity and anatomical size confers an advantage in detecting early nodal metastasis. While effective in detecting morphological changes, CT lacks metabolic data, limiting its sensitivity in early disease.

Avesani et al. (2020) retrospectively analyzed 297 ovarian cancer patients [13]. They found that CT-based Peritoneal Cancer Index (CT-PCI) correlated with residual disease post-surgery (OR = 1.04, *p* = 0.003), though its utility as a triage tool was limited (AUC = 0.64) [13]. Fagotti scoring applied to CT and PET/MRI effectively predicted resectability in our dataset. However, in one instance, a CT-based score of 8 and a PET/MRI score of 7, with anterior diaphragmatic adenopathy and receiving neoadjuvant chemotherapy, demonstrated the utility of PET/MR.

In our cohort, PET/MR and CT showed comparable accuracy for diaphragmatic involvement (78% vs. 74%). Hynninen et al. reported higher accuracy for PET/CT (78%) than CT (55%) in detecting diaphragmatic involvement [14]. PET/CT was superior to CT for detecting carcinomatosis in certain areas (accuracy of 78% vs. 55% for the diaphragm) [14]. Both modalities, however, demonstrated poor sensitivity for small bowel mesenteric disease (74% vs. 51%) in their study, slightly lower than our findings (PET/MR: 89%; CT: 79%). Additionally, PET/CT identified more cases of extra-abdominal metastasis than CT (32 vs. 25 patients). In our study, PET/MR detected extra-abdominal metastasis in 3 of 19 patients.

Tozzi et al. evaluated bowel involvement detection in stage IIIC–IV ovarian cancer using CT and exploratory laparoscopy (EXL), finding that CT alone had low sensitivity (56.7%) and specificity (72.4%). Combining CT with EXL improved sensitivity to 87.5% and accuracy to 79.6%, with EXL alone achieving the highest sensitivity (84.4%) and specificity (93.8%) [15]. The study concluded that CT alone has limited diagnostic power, but combining CT with EXL increases diagnostic accuracy for planning bowel resections. Onda et al. conducted a study on 147 patients with ovarian cancer to evaluate the accuracy of preoperative CT in diagnosing stage III disease [16]. In our research, when benchmarked against findings from exploratory laparotomy and debulking procedures, CT demonstrated a sensitivity of 100% and specificity of 89%.

In contrast, PET/MR showed a markedly lower sensitivity of 33% but comparable specificity of 89%. These results indicate that PET/MR is more effective in detecting metabolically active lesions, while CT provides superior identification of morphologically advanced metastases owing to its higher spatial resolution. Therefore, a combined imaging approach leveraging the metabolic sensitivity of PET/MR and the anatomical detail of CT may offer a more comprehensive and accurate assessment for preoperative staging and surgical planning in ovarian cancer.

## 5. Strengths and Weaknesses

This study comparing PET/MR and CT for detecting ovarian cancer metastases demonstrates several strengths and weaknesses. A key strength is its comprehensive approach, evaluating multiple anatomical sites and providing a nuanced understanding of each modality’s performance across different regions. The use of pathology as the gold standard ensures high diagnostic validity. This study highlights the complementary nature of PET/MR and CT, suggesting the potential benefits of a combined approach in clinical practice.

However, this study is limited by its small sample size of only 19 patients, which may compromise statistical power and generalizability. The inclusion criteria for patients with ovarian cancer or high suspicion of ovarian malignancy could introduce selection bias, potentially affecting the applicability of results to a broader patient population. Additionally, the study lacks information on inter-observer variability and does not include a cost-effectiveness analysis, an essential consideration for clinical implementation. The absence of long-term follow-up data also limits insights into the clinical impact of using these imaging modalities for treatment planning and patient outcomes. Also, this study did not stratify diagnostic performance by histological subtype, ascites volume, diagnostic SUV threshold, or diffusion-weighted MRI findings.

## 6. Implications for Practice and Future Research

The findings of this study have significant implications for clinical practice and future research in ovarian cancer imaging. The complementary strengths of PET/MR and CT suggest that a combined imaging approach could optimize the detection and characterization of metastatic sites. In practice, clinicians may consider using PET/MR for its superior sensitivity in detecting subtle metastases, particularly in areas like perihepatic disease and small bowel involvement. This dual-modality strategy could enhance the accuracy of preoperative staging and improve treatment planning. Future research should focus on developing standardized protocols for integrating PET/MR and CT in ovarian cancer evaluation and investigating the cost-effectiveness and impact of this combined approach on patient outcomes. Additionally, prospective studies with larger sample sizes are needed to validate these findings and further explore the potential of advanced image analysis techniques, such as artificial intelligence, to improve the diagnostic performance of these imaging modalities.

## 7. Conclusions

This study emphasizes the complementary diagnostic strengths of PET/MR and contrast-enhanced CT (CECT) in evaluating ovarian cancer metastases. PET/MR correctly identified all unresectable patients and demonstrated superior accuracy in detecting small bowel mesenteric disease, serosal involvement, and pelvic adenopathy. It also revealed metastases missed on CT, reflecting greater sensitivity in several regions. Conversely, CECT performed better in assessing colonic serosal invasion. Both modalities, however, exhibited discrepancies when compared with exploratory laparotomy findings.

While PET/MR offers advantages in functional imaging and helps minimize overdiagnosis, it has a better specificity and NPV; CT has a better sensitivity. Integrating both modalities can enhance preoperative staging accuracy and guide optimal treatment planning. Due to the study’s small sample size, preliminary findings, and larger, prospective studies, they are warranted to validate these findings and further explore with larger prospective trials.

## Figures and Tables

**Figure 1 cancers-17-02612-f001:**
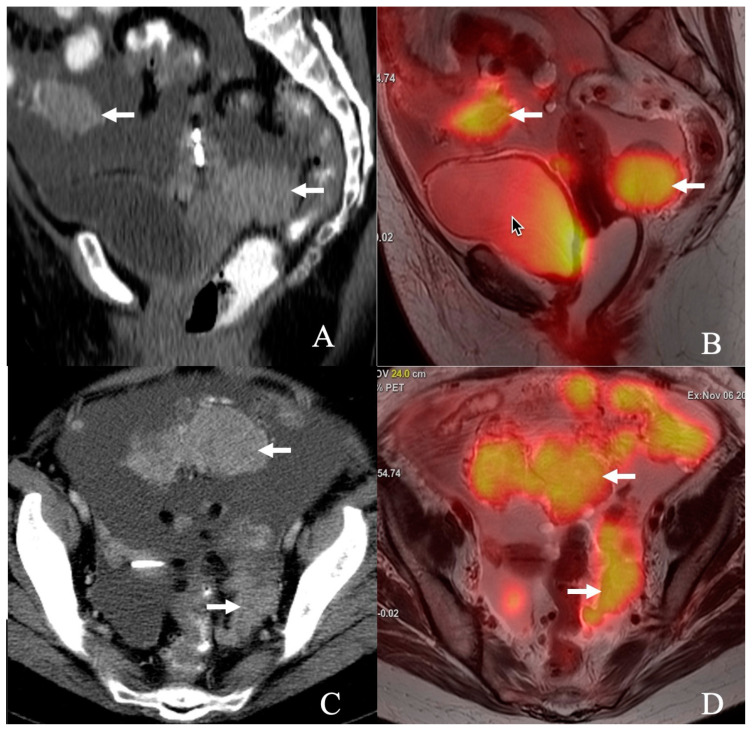
A 68-year-old female presents with adenocarcinoma, Mullerian primary/high-grade serous-type ovarian carcinoma with peritoneal carcinomatosis. (**A**) Sagittal, (**C**) axial contrast-enhanced CT, and (**B**) sagittal, (**D**) axial T2-weighted PET/MR images show FDG-avid peritoneal masses (arrow) in the pelvis and diffuse ascites.

**Table 1 cancers-17-02612-t001:** Fagotti score for metastases based on CT and PET/MR baseline imaging.

PET/MR Score	Study CT Score	Neoadjuvant Chemotherapy	Reason
4	3		
0	1		
11	10	Yes *	Extensive disease, supraclavicular adenopathy on PET/MR
11	8	Yes	Extensive disease
0	0		
5	4		
10	8	Yes	Extensive disease
7	8	Yes *	Anterior diaphragmatic adenopathy on PET/MR
5	5		
0	0		
9	8	Yes	Extensive disease
6	7		
1	2		
5	6		
0	2		
4	1		
3	2		
4	3	Yes *	Supraclavicular adenopathy on PET/MR
2	2		

* Patients obtained a biopsy of the nodes.

**Table 2 cancers-17-02612-t002:** Summary of metastases detected by PETMR and CT (Overall N = 19).

Site	PET/MR		CT	
	Absent (n, %)	Present (n, %)	Absent (n, %)	Present (n, %)
**Supraclavicular Adenopathy**	16 (88.9%)	2 (11.1%)	18 (94.7%)	1 (5.3%)
**Chest Adenopathy**	17 (89.5%)	2 (10.5%)	19 (100.0%)	
**Lung Nodules**	19 (100.0%)		19 (100.0%)	
**Pleural Effusion**	16 (84.2%)	3 (15.8%)	18 (94.7%)	1 (5.3%)
**Diaphragmatic Involvement**	17 (89.5%)	2 (10.5%)	11 (57.9%)	8 (42.1%)
**Perihepatic Disease**	14 (73.7%)	5 (26.3%)	12 (63.2%)	7 (36.8%)
**Hepatic Metastases**	19 (100.0%)		19 (100%)	
**Perisplenic Metastases**	16 (84.2%)	3 (15.8%)	14 (73.7%)	5 (26.3%)
**Supracolic Omental Disease**	12 (63.2%)	7 (36.8%)	9 (47.4%)	10 (52.6%)
**Infracolic Omental Disease**	11 (57.9%)	8 (42.1%)	2 (10.5%)	17 (89.5%)
**Retroperitoneal Adenopathy**	16 (84.2%)	3 (15.8%)	16 (84.2%)	3 (15.8%)
**Small Bowel Mesenteric Disease**	18 (94.7%)	1 (5.3%)	12 (63.2%)	7 (36.8%)
**Small Bowel Involvement**	17 (89.5%)	2 (10.5%)	17 (89.5%)	2 (10.5%)
**Colon Involvement**	14 (73.7%)	5 (26.3%)	13 (68.4%)	6 (31.6%)
**Pelvic Adenopathy**	17 (89.5%)	2 (10.5%)	18 (94.7%)	1 (5.3%)

**Table 3 cancers-17-02612-t003:** Diagnostic performance metrics for PET and CT.

Site	Statistic	PETMR	CT
**Perihepatic Disease**	Sensitivity	33%	67%
	Specificity	75%	69%
	Accuracy	68%	68%
	Positive Predictive Value	20%	29%
	Negative Predictive Value	86%	92%
**Small Bowel Mesenteric Disease**	Sensitivity	33%	100%
	Specificity	100%	75%
	Accuracy	89%	79%
	Positive Predictive Value	100%	43%
	Negative Predictive Value	89%	100%
**Small Bowel Involvement**	Sensitivity	100%	0%
	Specificity	94%	89%
	Accuracy	95%	84%
	Positive Predictive Value	50%	-
	Negative Predictive Value	100%	94%
**Colon Involvement**	Sensitivity	33%	100%
	Specificity	75%	81%
	Accuracy	68%	84%
	Positive Predictive Value	20%	50%
	Negative Predictive Value	86%	100%
**Pelvic Adenopathy**	Sensitivity	75%	25%
	Specificity	100%	87%
	Accuracy	95%	74%
	Positive Predictive Value	100%	33%
	Negative Predictive Value	94%	81%
**Diaphragmatic involvement**	Sensitivity	33%	71%
	Specificity	100%	75%
	Accuracy	78%	74%
	Positive Predictive Value	100%	63%
	Negative Predictive Value	75%	82%
**Omentum**	Sensitivity	85.71%	100.00%
	Specificity	83.33%	16.67%
	Accuracy	84.21%	47.37%
	Positive Predictive Value	75.00%	41.18%
	Negative Predictive Value	90.91%	100.00%

**Table 4 cancers-17-02612-t004:** Table outlining the criteria for determining resectability/non-resectability based on CT and PET/MR data.

Site/Criteria	CT Criteria for Non-Resectability	PET/MR Criteria for Non-Resectability	Results
Small bowel mesentery involvement	>1 site, >2 cm implants	FDG-avid lesions, >2 cm, DWI-positive	PET/MR higher specificity
Hepatic surface/parenchymal disease	>1 cm implant, multifocal	Metabolic activity + anatomical size	PET/MR better for small lesions
Lymph node metastasis	>1 cm short axis	FDG-avid regardless of size	PET/MR higher sensitivity
Colon serosal involvement	>2 cm implant	Metabolic activity, DWI-positive	CT higher sensitivity
Diaphragmatic involvement	>2 cm implant, thickening	FDG-avid, DWI-positive	Comparable accuracy
Fagotti score	>8	>8	Used for both modalities

## Data Availability

The original contributions presented in this study are included in the article. Further inquiries can be directed to the corresponding author.

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
