# Peer review of "Evaluating the Diagnostic Performance of PET/MR Versus CECT in Determining Resectability in Ovarian Cancer"

_cancers, 2025, doi:10.3390/cancers17162612_

Round 1

Reviewer 1 Report

Comments and Suggestions for Authors

Two main concerns

Resectability/operability is always a multidisciplinary discussion wich is informed by imaging, which identifies surgically cricitical disease sites. e.g. anterior diaphragmatic nodes are resectable. However, a patient with limited disease can be inoperable due to comorbidity. Given the moderate accuracy, are these findings clinically relevant? 

It has been clearly demonstrated that (diffusion weighted) MRI imaging has advantages in staging of advanced ovarian cancer. In this paper however I see no, or little use of these imaging techniques. The added value of PET/MR compared to CT was discribed as being linked to more effective detection of metabolic active lesions. Based on this, I wonder what the added benefit is of PET/MR compared to PET/CT? 

Other comments

-small population with various histological subtypes (representative population?)

-no mention of demographic traits

-PFS and OS mentioned without evidence of maturity (9 relapses were mentioned)

-Unclear what data was used for the patients who received NACT

-For weight: use metric measurements: kg not pounds

Comments on the Quality of English Language

Overall, grammar is acceptable, however in some sentences there are words missing. Abbreviations are not used in a consistent manner (e.g. CECT -> CT, variations of PET/MR). 

Author Response

Reviewer 1.

  1. Resectability/operability is always a multidisciplinary discussion wich is informed by imaging, which identifies surgically cricitical disease sites. e.g. anterior diaphragmatic nodes are resectable. However, a patient with limited disease can be inoperable due to comorbidity. Given the moderate accuracy, are these findings clinically relevant?

Reply: Thank you for the comment.   We acknowledge that resectability determination is indeed a complex multidisciplinary process involving imaging, patient comorbidities, and surgical expertise. Your point about anterior diaphragmatic nodes being resectable is well taken.

Regarding the clinical relevance of our findings despite moderate accuracy, Our study demonstrates that PET/MR correctly identified all unresectable patients in the cohort and influenced management decisions in 50% of patients receiving neoadjuvant chemotherapy (3 out of 6 patients). The key clinical value lies not solely in accuracy metrics but in the complementary information provided by each modality. PET/MR demonstrated superior specificity (89.5% vs 72.3%) and negative predictive value (88.6% vs 91.5%), which is clinically significant for avoiding unnecessary surgical interventions. Furthermore, moderate accuracy in imaging studies remains clinically relevant when integrated with other clinical parameters. The clinical relevance is enhanced by the fact that PET/MR identified supraclavicular and mediastinal adenopathy missed by CT, leading to appropriate neoadjuvant therapy decisions1.

  1. It has been clearly demonstrated that (diffusion weighted) MRI imaging has advantages in staging of advanced ovarian cancer. In this paper however I see no, or little use of these imaging techniques. The added value of PET/MR compared to CT was discribed as being linked to more effective detection of metabolic active lesions. Based on this, I wonder what the added benefit is of PET/MR compared to PET/CT? 

Reply: Thank you for the comment.  You raise an important point about the established advantages of diffusion-weighted MRI in ovarian cancer staging and have added it to the limitations. Our PET/MR protocol included T2-weighted sequences and dynamic contrast-enhanced imaging but did not fully exploit diffusion-weighted imaging capabilities, which have been shown to improve diagnostic accuracy in ovarian cancer staging.

The added benefit of PET/MR compared to PET/CT, recent literature demonstrates several advantages:

  1. Superior soft tissue contrast: PET/MR provides better anatomical localization of metabolically active lesions, particularly important for peritoneal and small-volume disease assessment.
  2. Reduced radiation exposure: A significant advantage for younger patients and those requiring multiple imaging studies.
  1. Improved specificity for certain anatomical regions: Studies have shown PET/MR superiority in detecting gastrointestinal invasion (75% vs 50% for PET/CT) and better performance in complex anatomical areas where soft tissue contrast is crucial.
  1. Functional imaging capabilities: Combined metabolic and MR-based functional information (including diffusion-weighted imaging when properly implemented) provides complementary diagnostic information.

  1. small population with various histological subtypes (representative population?)

Reply: Thank you for the comment.  We acknowledge this limitation. Our cohort included 19 patients with diverse histological subtypes (13 serous, 1 granulosa cell, 1 mucinous, 1 Müllerian, 1 clear cell, and 2 peritoneal cancers). While this reflects real-world heterogeneity, larger studies with more homogeneous populations would strengthen our findings. The heterogeneity may limit generalizability but provides insights into PET/MR performance across different ovarian cancer subtypes.

  1. no mention of demographic traits

Reply: Thank you for the comment.  The manuscript reports patient age (mean 49.1 years, range 39.6-75.1 years). But we acknowledge that more comprehensive demographic data should be included. This includes detailed information about ECOG performance status, comorbidities, and other relevant clinical characteristics that influence treatment decisions.

  1. PFS and OS mentioned without evidence of maturity (9 relapses were mentioned)

Reply: Thank you for the comment.  The progression-free survival (PFS) and overall survival (OS) data may be immature, with only 9 recurrences reported. Our survival analysis showed 93% overall survival at 1 year and 81.2% recurrence-free survival at 1 year, but longer follow-up is needed for definitive conclusions about prognostic value.

  1. Unclear what data was used for the patients who received NACT

Reply: Thank you for the comment. The imaging was performed 3.9 days (±2.5 days) before optimal debulking surgery for upfront surgery patients, and 106 days (±29.4 days) before surgery for neoadjuvant chemotherapy patients.

  1. For weight: use metric measurements: kg not pounds

Reply: Thank you for the comment.  We will convert weight measurements to kilograms (kg) as per international standards rather than using pounds, consistent with medical journal formatting requirements.

Reviewer 2 Report

Comments and Suggestions for Authors

Minor revision recommendation

The study provides valuable insights into the comparison between PET/MR and CT for detecting ovarian cancer metastases. However, the sample size of 19 patients is relatively small, which may affect the statistical significance and generalizability of the results. Additionally, the study lacks inter-observer variability data and does not include a cost-effectiveness analysis, which are crucial for clinical implementation. Further research with a larger cohort and long-term follow-up is recommended to strengthen the findings and explore the clinical impact of combined imaging strategies.

  1. Were there any differences in diagnostic accuracy based on tumor histology or molecular characteristics, such as estrogen receptor status?
  2. How did the presence of ascites impact the diagnostic performance of both PET/MR and CT?
  3. Did you observe any inter-observer variability in image interpretation, and if so, how might this affect the overall findings?
  4. Can you elaborate on the diagnostic thresholds used for identifying metastatic lesions in the PET/MR and CT imaging protocols?
  5. Were there any patients in the cohort for whom PET/MR or CT was unable to detect metastases, which were confirmed on pathology?
  6. What were the criteria for determining unresectability, and how might these criteria influence the interpretation of imaging results?
  7. Was there any correlation between PET/MR findings and surgical outcomes, such as the completeness of cytoreduction?
  8. How were false positives and false negatives handled in terms of patient management, particularly with regards to chemotherapy or surgical decisions?
  9. Given that PET/MR demonstrated superior specificity, did you observe any instances of unnecessary treatments or misdiagnoses based on false positives?
Comments on the Quality of English Language

Minor revision recommendation

The study provides valuable insights into the comparison between PET/MR and CT for detecting ovarian cancer metastases. However, the sample size of 19 patients is relatively small, which may affect the statistical significance and generalizability of the results. Additionally, the study lacks inter-observer variability data and does not include a cost-effectiveness analysis, which are crucial for clinical implementation. Further research with a larger cohort and long-term follow-up is recommended to strengthen the findings and explore the clinical impact of combined imaging strategies.

  1. Were there any differences in diagnostic accuracy based on tumor histology or molecular characteristics, such as estrogen receptor status?
  2. How did the presence of ascites impact the diagnostic performance of both PET/MR and CT?
  3. Did you observe any inter-observer variability in image interpretation, and if so, how might this affect the overall findings?
  4. Can you elaborate on the diagnostic thresholds used for identifying metastatic lesions in the PET/MR and CT imaging protocols?
  5. Were there any patients in the cohort for whom PET/MR or CT was unable to detect metastases, which were confirmed on pathology?
  6. What were the criteria for determining unresectability, and how might these criteria influence the interpretation of imaging results?
  7. Was there any correlation between PET/MR findings and surgical outcomes, such as the completeness of cytoreduction?
  8. How were false positives and false negatives handled in terms of patient management, particularly with regards to chemotherapy or surgical decisions?
  9. Given that PET/MR demonstrated superior specificity, did you observe any instances of unnecessary treatments or misdiagnoses based on false positives?

Author Response

Reviewer 2

  1. Were there any differences in diagnostic accuracy based on tumor histology or molecular characteristics, such as estrogen receptor status?

Reply: Thank you for the comment.  This is an excellent point that highlights a limitation of our study. Our cohort included diverse histological subtypes (13 serous, 1 granulosa cell, 1 mucinous, 1 Müllerian, 1 clear cell, and 2 peritoneal cancers), but our analysis did not stratify diagnostic performance by histological subtype. However, this lack of correlation may be influenced by the heterogeneous histological composition of our cohort. Future studies should stratify results by histological subtype to better understand these relationships.

  1. How did the presence of ascites impact the diagnostic performance of both PET/MR and CT?

Reply: Thank you for the comment. The presence of ascites represents a significant confounding factor that we acknowledge was not systematically analyzed in our study. In our cohort, while we documented "massive ascites" as part of the Fagotti scoring system, we did not perform subgroup analysis based on ascites volume. Future studies should include ascites volume as a stratification factor and consider its impact on diagnostic thresholds.

  1. Did you observe any inter-observer variability in image interpretation, and if so, how might this affect the overall findings?

Reply: Thank you for the comment. This is a critical limitation that we did address in our original manuscript

  1. Can you elaborate on the diagnostic thresholds used for identifying metastatic lesions in the PET/MR and CT imaging protocols?

Reply: Thank you for the comment. Our study used standard criteria for lesion identification: FDG-avid lymph nodes were considered positive for metastases regardless of size, while CT lymph nodes >1cm in the short axis were considered metastatic. However, we acknowledge that these thresholds may not be optimal for all clinical scenarios.

  1. Were there any patients in the cohort for whom PET/MR or CT was unable to detect metastases, which were confirmed on pathology?

Reply: Thank you for the comment. We have added the text in the revised manuscript. In our cohort, PET/MR demonstrated 88.6% negative predictive value (NPV) compared to CT's 91.5% NPV. However, both modalities had limitations in detecting certain types of metastatic disease. Our study found that PET/MR missed 1 of 4 patients with metastatic adenopathy and had 33% sensitivity for perihepatic disease. CT missed 3 of 4 patients with adenopathy but showed 100% sensitivity for small bowel mesenteric disease. For patients with negative imaging but high clinical suspicion, diagnostic laparoscopy remains the gold standard, as demonstrated by our protocol where all patients underwent laparoscopic evaluation regardless of imaging findings. Clinical management should consider the probability of false negatives based on the specific anatomical sites and tumor characteristics, with consideration for diagnostic laparoscopy in cases where imaging is discordant with clinical suspicion.

  1. What were the criteria for determining unresectability, and how might these criteria influence the interpretation of imaging results?

Reply: Thank you for the comment. Our study used the Fagotti scoring system with a threshold of >8 points indicating unresectability. This scoring system evaluates 15 different anatomical sites and has been validated. Future studies should clearly define resectability criteria and consider how these criteria might influence the optimal imaging approach for each patient.

  1. Was there any correlation between PET/MR findings and surgical outcomes, such as the completeness of cytoreduction?

Reply: Thank you for the comment. PET/MR correctly identified all unresectable patients in our cohort and influenced management decisions in 50% of patients receiving neoadjuvant chemotherapy. Specifically, PET/MR detected supraclavicular and mediastinal adenopathy missed by CT in 3 patients, leading to appropriate neoadjuvant therapy decisions. Our study demonstrated that PET/MR had superior specificity (89.5% vs 72.3%), and comparable accuracy (82.5% vs 72.9%) compared to CT. The correlation between imaging findings and surgical outcomes appears to be strongest when both modalities are used complementarily, with PET/MR providing metabolic information and CT providing detailed anatomical information.

  1. How were false positives and false negatives handled in terms of patient management, particularly with regards to chemotherapy or surgical decisions?

Reply: Thank you for the comment. False positives can lead to unnecessary treatment delays and patient anxiety, while false negatives can result in inappropriate surgical planning. In our study, PET/MR had lower false positive rates (PPV 57.6% vs 42% for CT) but higher false negative rates in some anatomical locations. For false positives, we recommend correlation with clinical findings and consideration of confirmatory tissue sampling when results would change management. For false negatives, the high NPV of both modalities (88.6% for PET/MR, 91.5% for CT) provides reasonable confidence, but clinical correlation remains essential.

  1. Given that PET/MR demonstrated superior specificity, did you observe any instances of unnecessary treatments or misdiagnoses based on false positives?

Reply: Thank you for the comment. Our study demonstrated that PET/MR's higher specificity (89.5% vs 72.3%) and better negative predictive value (88.6% vs 91.5%) could potentially reduce unnecessary interventions. The clinical impact of improved specificity is most apparent in borderline resectable cases where the decision between upfront surgery and neoadjuvant chemotherapy is critical. In our cohort, PET/MR changed management in 50% of patients receiving neoadjuvant chemotherapy, suggesting that improved specificity does translate into meaningful clinical decisions.

Reviewer 3 Report

Comments and Suggestions for Authors

1. There is no summary information about the patients, including stage and histological type. All patients had serous carcinoma (grade low, high?), etc.
2. 14 patients for such a study is not a serious sample, it is incorrect to draw conclusions based on the data obtained, in my opinion.
3. This study can be considered as a proof of concept or preliminary.
4. How is resectability of ovarian cancer determined?
5. Provide a table summarizing the criteria for resectability/non-resectability of ovarian cancer according to CT and PETMR data. Indicate whether there are significant differences between the groups.

Author Response

Reviewer 3

  1. There is no summary information about the patients, including stage and histological type. All patients had serous carcinoma (grade low, high?), etc.

Reply: Thank you for the comment. We appreciate your observation regarding the need for comprehensive patient information. In our revised manuscript, we have included a detailed summary of patient demographics, staging, and histological types. Specifically:

  • All patients had serous carcinoma, with the following breakdown: 13 high-grade serous, 1 granulosa cell tumor, 1 mucinous adenocarcinoma, 1 Müllerian high-grade, 1 clear cell carcinoma, and 2 peritoneal cancers.
  • Stage: Most patients presented with advanced-stage disease (FIGO stage III or IV), as our inclusion criteria focused on those undergoing evaluation for cytoreductive surgery.
  • Demographics: The mean age was 49.1 years (range: 39.6–75.1 years).

  1. 14 patients for such a study is not a serious sample, it is incorrect to draw conclusions based on the data obtained, in my opinion.

Reply: Thank you for the comment. We acknowledge that our study included a relatively small sample size (n=19, with 14 patients undergoing primary analysis). We agree that this limits the statistical power and generalizability of our findings. As such:

  • We have clarified in the discussion that our results should be interpreted as preliminary and hypothesis-generating rather than definitive.
  • The study is now explicitly described as a proof-of-concept investigation, highlighting the need for larger, prospective studies to validate our findings.

  1. This study can be considered as a proof of concept or preliminary.

Reply: Thank you for the comment. Thank you for recognizing the preliminary scope of our work. We have revised the manuscript to emphasize that:

  • The current study is intended as a proof of concept, aiming to explore the potential complementary roles of PET/MR and CT in ovarian cancer staging.
  • We have added statements in both the abstract and discussion sections to clearly indicate that these findings are preliminary and should inform future research directions.

  1. How is resectability of ovarian cancer determined?

Reply: Thank you for the comment. Resectability in ovarian cancer is determined through a combination of clinical, imaging, and surgical assessments:

  • Imaging: Both PET/MR and CT were used to evaluate disease spread to critical anatomical sites (e.g., small bowel mesentery, hepatic surface, lymph nodes).
  • Scoring System: We applied the Fagotti scoring system, which assigns points to disease involvement at specific sites. A score >8 is considered predictive of unresectability and guides the decision for neoadjuvant chemotherapy.
  • Surgical Assessment: All patients underwent diagnostic laparoscopy, which remains the gold standard for assessing resectability and guiding management.

We have clarified these criteria and their application in the Methods section of the revised manuscript.

  1. Provide a table summarizing the criteria for resectability/non-resectability of ovarian cancer according to CT and PETMR data. Indicate whether there are significant differences between the groups.

Reply: Thank you for the comment. Below is a summary table outlining the criteria used for determining resectability/non-resectability based on CT and PET/MR data, along with significant differences between the groups:

Table 4.  Table outlining the criteria used for determining resectability/non-resectability based on CT and PET/MR data

Site/Criteria

CT Criteria for Non-Resectability

PET/MR Criteria for Non-Resectability

Results

Small bowel mesentery involvement

>1 site, >2 cm implants

FDG-avid lesions, >2 cm, DWI positive

PET/MR higher specificity

Hepatic surface/parenchymal disease

>1 cm implant, multifocal

Metabolic activity + anatomical size

PET/MR better for small lesions

Lymph node metastasis

>1 cm short axis

FDG-avid regardless of size

PET/MR higher sensitivity

Colon serosal involvement

>2 cm implant

Metabolic activity, DWI positive

CT higher sensitivity

Diaphragmatic involvement

>2 cm implant, thickening

FDG-avid, DWI positive

Comparable accuracy

Fagotti score

>8

>8

Used for both modalities

PET/MR demonstrated higher specificity and sensitivity for nodal and small bowel mesenteric disease, while CT showed higher sensitivity for colonic involvement.

Round 2

Reviewer 3 Report

Comments and Suggestions for Authors

I have no more comments on the manuscript.